# Gaze and Evaluative Behavior of Patients with Borderline Personality Disorder in an Affective Priming Task

**DOI:** 10.3390/bs15091268

**Published:** 2025-09-17

**Authors:** Taavi Wenk, Michele Bartusch, Carolin Webelhorst, Anette Kersting, Charlott Maria Bodenschatz, Thomas Suslow

**Affiliations:** Department of Psychosomatic Medicine and Psychotherapy, University of Leipzig Medical Center, 04103 Leipzig, Germany; taavi.wenk@medizin.uni-leipzig.de (T.W.); michele.bartusch@medizin.uni-leipzig.de (M.B.); carolin.webelhorst@medizin.uni-leipzig.de (C.W.); anette.kersting@medizin.uni-leipzig.de (A.K.); charlott.bodenschatz@medizin.uni-leipzig.de (C.M.B.)

**Keywords:** borderline personality disorder, eye-tracking, affective priming, automatic emotion processing, facial expression, visual attention, negative interpretation bias, eye region

## Abstract

Borderline personality disorder (BPD) is associated with alterations in emotion processing. To date, no study has tested automatic emotion perception under conditions of unawareness of emotion stimuli. We administered a priming paradigm based on facial expressions and measured judgmental and gaze behavior during an evaluation task. A total of 31 patients with BPD and 31 non-patients (NPs) viewed a briefly shown emotional (angry, fearful, sad, or happy) or neutral face followed by a neutral facial expression (target). Areas of interest (AOI) were the eyes and the mouth. All participants included in our analysis were subjectively unaware of the emotional primes. Concerning evaluative ratings, no prime effects were observed. For early dwell time, a significant interaction between prime category and AOI was found. Both BPD patients and NPs dwelled longer on the eyes after the presentation of angry and fearful primes than of happy primes and dwelled longer on the mouth after the presentation of happy primes than of sad and neutral primes. Patients rated target faces more negatively. BPD patients, when compared to NPs, seem not to show alterations in automatic attention orienting due to covert facial emotions. Regardless of primes, individuals with BPD seem to be characterized by an increased negative interpretation of neutral facial expressions.

## 1. Introduction

Borderline personality disorder (BPD) is a severe mental disorder with a considerable lifetime prevalence in general population samples (up to 2.7% in the United States; [48]) and is associated with significant and stable functional impairment ([65]; [81]). The main symptom clusters are disturbances in relatedness, behavioral dysregulation, and affective dysregulation ([76]). [60]’s ([60]) biosocial model of BPD postulates that individuals with BPD are emotionally vulnerable due to three key factors: heightened emotional sensitivity (i.e., low threshold for recognition of emotional stimuli), greater emotional reactivity (i.e., high amplitude of emotional response), and a slow return to baseline (i.e., long duration of emotional response) (see also [21]).

To examine emotional sensitivity in BPD patients, facial expressions have been used in various neuroimaging investigations ([90]). BPD patients exhibited greater amygdala activity during the presentation of emotional facial expressions than non-patients (NPs) ([28]; [64]), confirming the hypothesis of increased emotional sensitivity in BPD. Interestingly, BPD patients also showed greater activation in response to masked fearful and happy faces than controls in the thalamus ([22]), which, alongside the amygdala, is assumed to play a crucial role in the processing of unconsciously perceived emotional stimuli ([56]). Individuals with BPD have been found to be hypersensitive to feelings of rejection ([7]; [78]) and to be more accurate in detecting fearful faces ([62]). Moreover, BPD patients seem to be characterized by social impairments, with a heightened sensitivity to emotional stimuli ([61]; [98]) and a negative bias when interpreting facial stimuli ([5]; [15]; [26]). However, there is a dearth of research examining automatic emotion perception in BPD. Increased perceptual sensitivity to emotional information at an automatic processing level could be an important component of the heightened emotional sensitivity observed in BPD patients.

A suitable method to investigate automatic emotional processing is the affective (facial) priming paradigm (e.g., [66]; [67]). In this paradigm, emotional facial expressions (primes) are presented briefly for 50 ms or less. The evaluation of subsequently shown neutral facial expressions (targets) seems to be affected by the valence of the prime ([33]; [70]). Typically, neutral (or emotional) target faces are rated more positively when preceded by a happy facial prime compared to when preceded by a negative facial prime. According to a biopsychological model of affective priming, facial emotions activate subcortical structures, the amygdalae in the case of negative expression and the nucleus accumbens in the case of positive expression, which then alter the readiness of regions involved in perception and behavior to respond to subsequent stimuli ([96], [97]). Evaluative shifts based on positive facial expressions are positively correlated with nucleus accumbens activation, whereas evaluative shifts based on negative facial expressions are positively correlated with amygdala activation ([85]). The affective priming task has been repeatedly administered to investigate the automatic evaluation of, and neural reactivity to, facial emotions in clinical samples. In some studies, priming effects were observed on the evaluative behavioral level, while in other studies, effects were only determined for neurobiological parameters. For example, schizophrenia patients suffering from anhedonia or flattened affect expression were found to be characterized by evaluative shifts due to masked negative facial expressions, whereas NPs showed evaluative shifts due to masked positive facial expressions ([86]). Depressed patients exhibited potentiated amygdala reactivity to masked negative facial expressions along with a reduced responsiveness to masked positive facial expressions compared with NPs ([84]). Amygdala reactivity to masked negative faces was found to be associated with negative evaluative shifts in depressed patients ([23]).

There are only two affective priming studies on individuals with BPD. [29] ([29]) compared the evaluative ratings and response latencies of individuals with BPD to those of healthy subjects in an affective priming task using angry, happy, neutral, or no facial expression as primes followed by a neutral target face. The authors found no differences between groups concerning affective priming parameters. They interpreted their findings as contradicting the assumption of a general automatic emotional hypersensitivity in BPD. In a subsequent study, [30] ([30]) administered an affective priming paradigm with lexical stimuli and again found no difference between individuals with BPD and healthy subjects in affective priming indices. Consequently, the assumption of altered automatic processing of emotional stimuli in BPD could not be confirmed in priming studies. However, the above-mentioned affective priming studies on BPD did not assess participants’ prime awareness. As awareness of prime stimuli can prevent prime effects on evaluative behavior ([66]), it still remains an open question as to whether, under conditions of subjective unawareness, individuals with BPD may manifest heightened evaluative priming effects compared to healthy subjects.

In recent decades, eye-tracking technology has emerged as a valuable method for research on attention deployment in emotional disorders (e.g., [40]; [83]). As attention deployment and gaze direction are tightly coupled, eye-tracking technology provides a direct measure of attention allocation ([100]). Eye-tracking technology allows for an examination of early attention allocation processes ([17]), such as initial attention orientation towards emotional stimuli. Early gaze behavior is assumed to be controlled by automatic emotion processing ([69]). Emotional facial expressions (EFEs) signal specific feeling states, intentions, or action requests. Angry facial expressions indicate a potential direct threat ([24]; [95]), while fearful facial expressions signal an indirect threat or dangers in the environment ([94]). Happy facial expressions are signals of approval or approach ([68]), whereas sad facial expressions indicate that the expresser is in need of help ([63]). Although neutral faces, by definition, do not express a particular emotion, they typically are perceived as somewhat negative ([57]; [74]) or signaling dominance in men ([42]). The regions of facial expressions relevant for emotion recognition vary depending on the expressed emotion: whereas the mouth is most relevant for the identification of happiness ([77]), the eyes are equally or more relevant for the identification of fear and sadness ([32]; [77]). For expressions of anger, the eyes appear to represent the most diagnostically important facial area ([32]).

In a recent eye-tracking investigation, [11] ([11]) administered an affective priming task with emotional and neutral prime faces and neutral target faces in healthy controls. The authors found that the prime faces elicited early eye movements towards the diagnostically relevant regions of the face (i.e., towards the eyes after presentation of a fearful prime and towards the mouth after presentation of a happy prime). Examining the gaze behavior of patients with clinical depression and healthy controls in the same affective priming task. [12] ([12]) observed that both patients and healthy controls would fixate first on the eye region with a higher probability after the presentation of negative (fearful or sad) primes compared to a happy prime. These findings indicate that depressed patients manifest normal attentional orientation when perceiving fearful and sad facial expressions outside of conscious awareness. As BPD is characterized by emotional dysfunctions ([60]; [76]; [78]; [98]), it remains an important task to clarify whether there are impairments in emotion perception at an automatic processing level in BPD. To the best of our knowledge, no study has yet examined the gaze behavior of individuals with BPD using an affective priming task.

Previous eye-tracking studies taking into account gaze behavior towards emotional content in individuals with BPD either used emotion classification or free viewing paradigms ([8]; [9]; [14]; [15]; [79]; [92]). In only three previous eye-tracking studies were single faces presented as experimental stimuli with a brief stimulus presentation condition included ([8], [9]; [79]). The results of these investigations are the most relevant to the present study. [8] ([8]) utilized an emotion classification paradigm (with happy, angry, and fearful facial expressions) and compared the gaze behavior and amygdala activation of individuals with BPD and healthy controls. In this study, individuals with BPD showed more and faster initial fixation changes to the eyes of briefly presented angry faces, which was associated with increased amygdala activation. Moreover, [9] ([9]) observed in an emotion classification paradigm during the brief presentation of faces (150 ms) that individuals with BPD made more and faster initial fixations towards the eyes of neutral faces compared to healthy controls. However, in aggressive patients with BPD, an avoidance of the eyes of angry faces was observed during the long presentation condition (5000 ms). The authors interpreted this finding as evidence of initial hypervigilance for social threat followed by eye avoidance in aggressive BPD patients. [79] ([79]) also examined gaze behavior in an emotion classification task, comparing individuals with BPD to healthy controls. Individuals with BPD were found to direct their gaze more quickly towards the eyes of emotional and neutral faces in the brief presentation condition (150 ms), indicating a general early vigilance with respect to the eyes, the diagnostically relevant region of social threat. In contrast, in the long presentation condition (5000 ms), patients with BPD did not fixate on the eyes of emotional and neutral faces any longer than the healthy volunteers. In the study of [47] ([47]), morphed emotional faces had to be labeled. BPD patients showed a general tendency to fixate longer on the eye region—regardless of the emotional combinations the faces were displaying—than healthy controls. If one summarizes the results of previous eye-tracking studies with regard to attention paid to the eyes, it can be concluded that BPD patients, compared to healthy controls, seem to be characterized by an initial hypervigilance to the eyes of emotional ([79]) and neutral faces ([9]; [79]). In contrast, BPD patients appear not to manifest alterations in attention allocation to the eyes when longer time intervals are considered and clearly neutral or clearly emotional facial expressions are shown ([9]; [79]). After an initial moment of hypervigilance, aggressive BPD patients could be characterized by attentional avoidance of the eyes ([9]). However, when viewing emotionally ambiguous faces, BPD patients seem to look longer at the eyes than healthy controls ([47]). Patients may more intensively search the eye region for information to resolve the ambiguity of the expression. Patients with BPD do not, in general, seem to allocate their attention to the eyes of others for longer than NPs.

Utilizing a free-viewing paradigm, [14] ([14]) reported no alterations in early gaze behavior (i.e., time to first fixation) in response to socioemotional stimuli in individuals with BPD compared to healthy controls. Similarly, [92] ([92]) found no alterations in early gaze behavior towards EFEs in individuals with BPD compared to NPs in a free-viewing paradigm with EFEs. In contrast, [15] ([15]) observed increased first fixation latency for negative socioemotional pictures compared to healthy control subjects in a free-viewing paradigm. The two studies of [14] ([14], [15]) provided mixed results concerning early processes of attention allocation during the perception of socioemotional stimuli in BPD. None of the above-mentioned eye-tracking studies focused on the effect of masked facial emotions on attention processes in BPD.

The present study primarily aimed to investigate how individuals with BPD automatically process facial emotional information, without conscious awareness, compared to NPs. We focused our analysis on how gaze and evaluative behavior are influenced by masked facial emotions. We used eye-tracking technology to examine early (“probability of first fixation” and “early dwell time” (i.e., dwell time during the first 500 ms of target presentation)) and late (dwell time during the whole time of target presentation) gaze behavior in an affective priming task presenting angry, fearful, sad, and happy prime faces and assessed the evaluative ratings of neutral target faces as a function of the prime category. To the best of our knowledge, this study is the first eye-tracking investigation to examine the influence of masked facial emotions on gaze behavior and evaluative judgments in patients with borderline personality disorder (compared to NPs). The present study aims to contribute to filling a gap in the research on the automatic processing of socially relevant information in patients with borderline personality disorder. Further objectives of our study were to examine whether BPD patients manifest an early vigilance for the eyes and a negative bias when evaluating facial expressions.

Theoretical considerations ([21]; [60]) and previous findings in BPD research ([8]; [62]; [64]) led us to the assumption that individuals with BPD may be characterized by an increased emotional sensitivity to social threat on an automatic processing level. We therefore expected to find both altered gaze behavior and altered evaluative ratings due to threat-related facial emotions in BPD compared to NPs. However, it must be noted that previous findings on emotional sensitivity in BPD are not consistent and that there is also evidence of a higher general sensitivity to emotional information in BPD patients (e.g., [28]; [22]). The present priming study differs from the previous one by [29] ([29]) in that our study participants did not consciously perceive emotional primes. Considering the early attention parameters “probability of first fixation” and “early dwell time,” we expected that both groups would first fixate with a higher probability on the eyes of neutral target faces after the presentation of fearful, angry, and sad primes compared to that of happy and neutral primes. Similarly, both groups were expected to dwell longer on the eyes of neutral target faces after the presentation of fearful, angry, and sad primes compared to happy and neutral primes during the first 500 ms after prime presentation. These two hypotheses concerning the interaction effect of the task are based on the previous findings of eye-tracking studies in our laboratory in which the affective priming task was administered ([11], [12]). We further assumed that individuals with BPD would fixate first on the eye region of a neutral target face with a higher probability after angry and fearful prime faces than the NP group. We also assumed that individuals with BPD would dwell longer on the eye region of a neutral target face after angry and fearful prime faces than the NP group during the first 500 ms. Additionally, we hypothesized that, irrespective of prime category, individuals with BPD would fixate with a higher probability on the eyes first compared with the NP group (cf. [9]; [79]). Lastly, we expected individuals with BPD to dwell longer on the eye region of target faces than the NP group, irrespective of prime category (cf. [47]), and we expected the BPD group to evaluate neutral target faces more negatively in general, irrespective of the prime category (cf. [5]; [15]; [26]), and especially more so after the presentation of negative prime faces, compared to NPs.

Comorbid mental disorders are a common problem in BPD research. As only including individuals with BPD without a comorbid mental disorder would not represent the BPD population ([80]), we did not exclude BPD patients with one or multiple comorbid mental disorders from our study. We assessed trait anxiety, depressive symptom severity, and experiences of childhood maltreatment in our study participants, which are characteristics that occur more strongly or more frequently in patients with BPD than in NPs ([6]; [44]; [53]), and we examined their correlation with gaze behavior and evaluative ratings. Depressive mood and dispositional anxiety can influence the way faces are viewed and interpreted. For example, depressive symptoms are associated with less visual attention paid to other people’s eyes ([82]) and increased negative interpretation of neutral facial expressions ([93]). Trait anxiety has been found to be linked to attentional bias for threatening facial expressions ([49]) and to facilitate unconscious perceptual analysis of social threat stimuli ([58], [59]). In our study, the NPs did not have any mental health diagnoses.

## 2. Materials and Methods

### 2.1. Participants

All participants in this study were native German speakers aged between 18 and 44 years. Patients with BPD were recruited from the Department of Psychosomatic Medicine and Psychotherapy at the University of Leipzig. The senior physician of the department referred patients with BPD to our research group. The participants of the NP group were recruited via online advertisements and public notices posted in the city of Leipzig. The testing of individuals from both groups was conducted at the Department of Psychosomatic Medicine and Psychotherapy at the University of Leipzig.

General exclusion criteria were as follows: (1) current or lifetime neurological disorder; (2) head injury with a possible negative impact on cognitive function; (3) current substance dependence or substance abuse; (4) drug use on the day of the experiment; (5) current medication with benzodiazepines; and (6) compromised vision. We assessed cognitive flexibility (TMT-B, see the Measures section) and visual acuity (Snellen eye chart). All other exclusion criteria were assessed by self-report.

An exploratory a priori power analysis using G*Power (Version 3.1.9.6; [34]) was conducted to determine the minimum group size to detect small within-group priming effects in gaze data (cf. [12]). According to the results of our power analysis, a minimum total sample size of 60 participants was determined to be necessary to detect small within-group priming effects in “probability of first fixation” (η_p_^2^ = 0.05, *α =* 0.05), with a power of 80%. As no previous study has investigated the affective priming of gaze behavior in patients with BPD, it was not possible to perform a priori power analysis concerning between-group differences. However, we planned to exceed the calculated minimum total sample size and aimed for a group size of 50 individuals per group.

Exclusion criteria for patients with BPD included a diagnosis of bipolar disorder, psychotic disorder, or schizoaffective disorder. The presence of additional comorbid personality disorders was not an exclusion criterion in our study. We initially scheduled experimental sessions for 51 patients. However, due to COVID-19-related cancelations, only 49 patients were tested. We excluded the following 18 patients after data collection: (1) patients who did not achieve a minimum score of five out of nine DSM-IV BPD criteria (*n* = 4); (2) patients with prime awareness, as indicated by our awareness-test (see below) (*n* = 4); (3) those exceeding an eye-tracker calibration deviation of 1° on either the x- or the y-axis (*n* = 8); and (4) those with current substance abuse (*n* = 2). All patients were screened with SCID I and II ([36]; [99]) by two trained interviewers. The interviewers were a psychologist and a student trained by a psychologist. The final sample consisted of 31 (26 female) individuals with BPD and 31 (26 female) NPs. The individuals from the NP group were part of a larger sample and matched to the BPD group for age and gender. The exclusion criteria for the NP group were as follows: (1) current or lifetime diagnosis of any psychiatric disorder; (2) past or current use of psychotherapy; (3) minimal, moderate, or severe depression (BDI-II ≥ 9); and (4) exceeding the cut-off score for moderate BPD symptoms (BSL-23 score > 1.07; [51]). The Mini-International Neuropsychiatric Interview (M.I.N.I.; [1]) was used in the NP sample to check for exclusion criteria such as depression or any history of psychiatric disorders. All individuals from the NP group were additionally screened using SCID I and II. The descriptive statistics of psychological and socioeconomic measures for both groups are presented in Table 1. The frequency of medication intake and comorbid mental disorders in the BPD group is presented in Table 2. Prior to the commencement of this study, all participants provided written informed consent. Furthermore, they were financially compensated upon the completion of the study.

### 2.2. Measures

All study participants completed a sociodemographic questionnaire assessing age, biological sex, and education. Furthermore, depressive symptoms were assessed using the Beck Depression Inventory (BDI-II; German version: [43]). The BDI-II is a self-report questionnaire with 21 items. It assesses depressive symptoms such as negative cognitions, hopelessness, and physical symptoms during the preceding two weeks. In the present sample, the BDI-II had excellent internal consistency (α = 0.936). To assess participants’ dispositional anxiety, the State–Trait Anxiety Inventory (STAI; German version: [55]) was administered. The STAI comprises two distinct versions, one for the assessment of trait anxiety (STAI-T) and the other for the measurement of state anxiety (STAI-S). Each version has 20 items utilizing a 4-point Likert scale for responses. In the present sample, internal consistency was excellent (α = 0.964) for the STAI-T. Adverse childhood experiences were assessed using the Childhood Trauma Questionnaire (CTQ; German version: [4]). The CTQ assesses childhood trauma via retrospective self-report and consists of five subscales: emotional abuse, physical abuse, sexual abuse, emotional neglect, and physical neglect. In the present sample, the internal consistency of the CTQ (total score) was high (α = 0.898). Borderline symptom severity was assessed using the Borderline Symptom List (BSL-23; [13]). The BSL-23 is based on the DSM-IV and assesses the intensity of BPD symptoms over the previous week via 23 items on a 5-point Likert scale. The internal consistency in the present sample was excellent (α = 0.969) for the BSL-23. Part B of the Trail Making Test (TMT-B; [73]) was administered as a measure of cognitive flexibility. In this test, the participants’ task is to connect numbers and letters in ascending order. The total time needed for task completion serves as an indicator of cognitive flexibility.

### 2.3. Stimuli and Affective Priming Task

Trials consisted of 60 test trials (5 emotional prime conditions × 12 actors) and 5 practice trials. A test trial began with a fixation cross until a fixation of 1000 ms. A prime stimulus followed for 50 ms with forward and backward masking of 50 ms (see Figure 1). Subsequently, the neutral target face followed for 1850 ms. After the target stimulus, a gray screen appeared until the target face was evaluated verbally on a 4-point Likert scale (1 = very negative; 2 = slightly negative; 3 = slightly positive; 4 = very positive). The same procedure had been applied by [11] ([11], [12]). Five different EFEs served as prime stimuli: happiness, fear, anger, sadness, and neutrality (prime condition). The EFEs were clearly expressed by each of the actors. Each prime category was presented 12 times. Neutral faces served as target stimuli. Prime and target stimuli always displayed the same actor. To avoid showing a stimulus under the neutral prime condition that is identical to the neutral target, the stimulus photographs of neutral targets were vertically mirrored and used as neutral primes. The masks consisted of 79 randomly chosen letters presented on a gray background. Each emotion prime category was shown three times in a row to maximize priming effects. Furthermore, to control for potentially confounding prime sequence effects, the order of the prime categories was controlled by generating five different fixed randomization patterns (Latin square design). Within these randomization sequences, no actor was presented in two consecutive trials.

Facial stimuli were chosen from the Radboud Faces Database (RaFD: [54]). Prime and target stimuli in all trials were gray-scaled. Half of the actors were female. The display size of each facial expression was 23 cm high × 17 cm wide. Five photographs containing no main facial features (ears, nose, and eyes replaced by a gray surface) served as primes in five practice trials. After the practice trials, 60 experimental trials followed. The duration of the experiment was approximately 8 min.

### 2.4. Procedure and Eye-Tracking Experiment

The experiment was conducted as part of a series of experiments, all of which were performed in the same experimental session. The sequence of experiments was fixed, with the priming experiment occurring second. All experiments took place in the eye-tracking laboratory of the Department of Psychosomatic Medicine and Psychotherapy at the University of Leipzig. The procedure was explained before the experiment. Participants sat in front of a computer screen at a distance of approximately 70 cm. Ceiling illumination was stable. The Mavolux 5032B luxmeter (Gossen, Nuremberg, Germany) was used to measure both horizontal and vertical illumination. The horizontal illumination was measured by placing the sensor on the desk in front of the screen, whereas the vertical illumination was measured by placing the sensor at about the position where participants’ eyes were located. The horizontal illumination was 570 lx, and the vertical illumination was 250 lx. Before starting the experiments, camera adjustments were made to ensure the best image quality. Next, a calibration (five points) and separate validation (using four different points) procedure was administered. We did not examine drift correction. The mean calibration deviation was *M* = 0.53° (*SD* = 0.18) for the x-axis and *M* = 0.42° (*SD* = 0.19) on the y-axis in the NP group, and *M* = 0.48° (*SD* = 0.19) on the x-axis and *M* = 0.40° (*SD* = 0.21) on the y-axis in the BPD group. Both groups did not differ regarding the mean calibration deviation on either the x- or the y-axis (all *p*s ≥ 0.122; two-tailed). Subjects were instructed in written form on the screen to view a series of faces. They were not informed about the prime stimuli. They were asked to evaluate verbally the valence of the facial expression immediately after the target faces disappeared. After the eye-tracking experiment, participants were asked to fill out the BDI-II, STAI-T, CTQ, and the BSL-23 and to perform the TMT-B.

### 2.5. Awareness Check

The present study aimed to investigate the processing of emotional information without conscious awareness. Prior research indicates that when presenting a prime for 50 ms combined with forward and backward masking of 50 ms each, primes are processed subliminally (see [50]). Additionally, an awareness check was administered to assess awareness of emotional prime stimuli. The assessment procedure was identical to that used in our previous eye-tracking studies investigating affective priming effects ([11]; [12]): all participants were asked whether during the experiment they had noticed anything uncommon. In case participants replied that they had seen faces or affective expressions prior to the target faces, they were additionally asked whether they had seen happy, sad, angry, or fearful faces. Participants who identified at least one emotional quality were excluded from the data analysis.

### 2.6. Apparatus

Stimuli were presented on a 22-inch TFT widescreen monitor (resolution: 1680 × 1050). The experiment was conducted using a customized SensoMotoric Instruments (SMI) (Teltow, Germany) Dell laptop. The eye-tracking recording was conducted using an IView X RED250 remote system manufactured by SMI (Teltow, Germany). The IView X RED250 has a sample frequency of 250 Hz and a gaze position accuracy of 0.4° and compensates for head movements. Therefore, no chin rest was needed. SMI Experiment Center software (Version 3.4.119) was used for stimulus presentation and synchronization with recorded eye movements.

### 2.7. Eye Movement Parameters

SMI BeGaze (Version 3.4.27) was used to define areas of interest (AOIs) and to process eye-tracking data. Eye-tracking data was computed by a velocity-based algorithm. The minimum fixation duration was set at 100 ms, while the minimum saccade duration was set at 22 ms, with a peak velocity threshold of 40°/s (see also [12]; [92]). The surface of the eyes and the mouth of the target face defined two separate AOIs. The mouth region (*M* = 21,152 square pixels; *SD* = 2447) included the mouth and its surrounding area. The eye region (*M* = 49,918 square pixels, *SD* = 3616) included the eyes, the eyebrows, and the area in between (glabella). For all individuals in the photographs, AOIs were delineated in a comparable manner by selecting fixed reference points (10 for the eyes and 7 for the mouth) (see Figure 2). These reference points were consistently applied to each individual to maintain methodological uniformity. AOIs were manually defined by a single coder; to ensure consistency, the procedure was strictly standardized. “Dwell time” and “probability of first fixation” were used as dependent measures. “Dwell time” was used as an indicator of sustained attention allocation and was calculated by summing up all fixation and attentional shifts that remained in one of the AOIs (eyes or mouth) in milliseconds. It was calculated separately for each of the prime conditions (happy, fearful, angry, sad, and neutral) and AOIs by averaging for each of the prime conditions and AOIs separately across trials and participants. We used “probability of first fixation” as an indicator of early attention allocation. The calculation was performed by coding either 0 or 1, contingent on the location of the initial AOI fixation. Thereafter, it was summed up for both AOIs and all prime conditions separately and divided by the total number of trials. Thus, a probability score of 1 or 0 indicates that the initial fixation was on a specific AOI in a specific prime condition for all trials or for none of the trials, respectively. We used “early dwell time” as a second indicator of early attention allocation. To calculate “early dwell time,” the same method as for dwell time was used but restricted to the first 500 ms.

### 2.8. Statistical Analysis

For testing differences between groups on psychological and socioeconomic measures, independent-sample *t*-tests were administered. If the assumption of homogeneity of variances was violated, as assessed using Levene’s test, we adjusted the degrees of freedom accordingly. When variables were not normally distributed, we conducted additional Mann–Whitney U tests. As the non-parametric results did not differ from the corresponding t-tests, we report only the parametric results. We also compared individuals taking antidepressant medication with those not taking such medication. For non-normally distributed variables in this comparison, the Mann–Whitney U test was used instead.

Evaluative ratings were analyzed using a 5 (prime category: happy, fearful, angry, sad, neutral) × 2 (group: BPD, NP) mixed-model ANOVA. To evaluate the general interpretation biases in both groups, we further tested whether evaluative ratings differed from the center of the evaluation scale (2.5, which would indicate a neutral evaluation) using one-sample *t*-tests. Due to technical problems, evaluative data were not available for one patient. “Dwell time,” “probability of first fixation,” and “early dwell time” data were analyzed, examining a 5 (prime category: happy, fearful, angry, sad, neutral) × 2 (AOI: eyes, mouth) × 2 (group: BPD, NP) mixed-model ANOVA. In case of violation of sphericity, the [39] ([39]) correction was applied. If the assumption of normally distributed residuals was violated, we additionally performed Mann–Whitney U tests. As no result differed from the parametric analyses, only the latter are reported. When differences between conditions were not normally distributed, we additionally conducted a Wilcoxon signed-rank test and reported the corresponding results if they differed from the parametric outcomes. Since the randomization sequences were unequally distributed due to the exclusion of participants following data collection, we additionally performed one-way ANOVAs to investigate the influence of this potentially confounding factor on eye-tracking data and evaluative ratings for both groups separately.

Further explorative analyses were made using Pearson’s correlation coefficient. We correlated clinical and affect measures with eye-tracking data and evaluative ratings in both groups separately. If residuals were not normally distributed, we used Spearman’s rank correlation coefficient instead. Effect sizes are reported: Cohen’s *d* for *t*-tests, η_p_^2^ for mixed ANOVAs, *r* for Wilcoxon test, and *r* for Mann–Whitney-U tests. In all instances of multiple testing, the Benjamini–Hochberg (BH) correction of *p*-level was applied. The results of all statistical assumption tests conducted for the main analyses are presented in Table A6 of the Appendix B.

## 3. Results

### 3.1. Group Characteristics

Independent-sample *t*-tests indicated a significant difference for severity of depression (BDI-II; *t*(36.84) = −11.66, *p* < 0.001, two-tailed), borderline symptoms (BSL-23; *t*(32) = −11.76, *p* < 0.001, two-tailed), trait anxiety (STAI-T; *t*(60) = −13.18, *p* < 0.001, two-tailed), and childhood maltreatment (CTQ; *t*(40.25) = −8.44, *p* < 0.001, two-tailed). There was no significant difference in school years or cognitive flexibility (TMT-B; all *ps* > 0.05, two-tailed).

In the BPD group, there were differences concerning “dwell time,” “early dwell time,” and “probability of first fixation” between patients taking antidepressant medication vs. those not taking medication, while there were no differences in evaluative ratings (see Table A1, Table A2, Table A3 and Table A4 in the Appendix A).

### 3.2. Behavioral Data: Evaluative Ratings

Evaluative ratings are presented in Figure 3. The results of evaluative ratings indicated no main effect of prime (*F*(4, 236) = 1.33, *p* = 0.258, η_p_^2^ = 0.02) and no interaction effect between prime and group (*F*(4, 236) = 1.73, *p* = 0.150, η_p_^2^ = 0.03). The main effect of group was significant (*F*(1, 59) = 4.98, *p* = 0.029, η_p_^2^ = 0.08). The evaluation of target faces was more negative in the BPD group than in the NP group. One-way ANOVAs using a randomization sequence as a between-subject factor indicated no effect of the randomization sequence on evaluative ratings (all *p*s > 0.132). The results of one-sample *t*-tests indicated that the BPD group significantly differed in its evaluation of the neutral target faces after happy (*t*(29) = −6.98, *p* < 0.001 (two-tailed), *d* = −1.27, CI [−1.75, −0.78]), fearful (*t*(29) = −6.71, *p* < 0.001 (two-tailed) *d* = −1.22, CI [−1.69, −0.74]), angry (*t*(29) = −6.44, *p* < 0.001 (two-tailed), *d* = −1.18, CI [−1.64, −0.70]), sad (*t*(29) = −7.54, *p* < 0.001 (two-tailed), *d* = −1.38, CI [−1.87, −0.87]), and neutral faces (*t*(29) = −5.60, *p* < 0.001 (two-tailed), *d* = −1.02, CI [−1.46, −0.57]) from the center of the scale (i.e., 2.5). We found similar results for the NP group, indicating that the evaluations of neutral target faces after happy (*t*(30) = −3.68, *p* < 0.001 (two-tailed), *d* = −0.66, CI [−1.05, −0.27]), fearful (*t*(30) = −3.76, *p* < 0.001 (two-tailed), *d* = −0.68, CI [−1.06, −0.28]), angry (*t*(30) = −4.08, *p* < 0.001 (two-tailed), *d* = −0.73, CI [−1.13, −0.33]), sad (*t*(30) = −2.89, *p* = 0.007 (two-tailed), *d* = −0.52, CI [−0.89, −0.14]), and neutral primes (*t*(30) = −2.78, *p* = 0.009 (two-tailed), *d* = −0.50, CI [−0.87, −0.12]) also differed from the center of the scale. Thus, both groups evaluated neutral target faces as negative, irrespective of prime category.

### 3.3. Eye-Tracking Data: Probability of First Fixation, Early Dwell Time, and Dwell Time

Table 3 and Table 4 provide the descriptive statistics of the eye-tracking data. Analysis of the “probability of first fixation” revealed a significant main effect of AOI (*F*(1, 240) = 21.05, *p* < 0.001, η_p_^2^ = 0.26) but no significant main effect of prime (*F*(3.30, 219.17) = 0.98, *p* = 0.418, η_p_^2^ = 0.02) or group (*F*(1, 60) = 1.27, *p* = 0.265, η_p_^2^ = 0.02). The prime × group interaction effect (*F*(3.30, 219.17) = 0.24, *p* = 0.884, η_p_^2^ < 0.01), the AOI × group interaction effect (*F*(1, 240) = 0.73, *p* = 0.395, η_p_^2^ = 0.01), and the prime × AOI interaction effect (*F*(4, 240) = 1.03, *p* = 0.394, η_p_^2^ = 0.02) were not significant. The prime × AOI × group interaction effect was also not significant (*F*(3.65, 219.17) = 0.55, *p* = 0.685, η_p_^2^ = 0.01).

As residuals were not normally distributed, we additionally performed non-parametric single comparisons for independent samples to test for group differences. The results confirmed that the groups did not differ on either of the parameters for “probability of first fixation” (all *p*s > 0.112). To account for the potentially confounding factor of the randomization sequence, we performed a non-parametric one-way ANOVA. The results indicated no effect of the randomization sequence (all *p*s > 0.373).

“Early dwell time” data is presented in Figure 4 and Figure 5. Analysis of “early dwell time” data revealed no main effect of the prime (*F*(4, 240) = 1.61, *p* = 0.172, η_p_^2^ = 0.03) but a significant main effect of the AOI (*F*(1, 240) = 16.58, *p* < 0.001, η_p_^2^ = 0.22), and no significant main effect of group (*F*(1, 60) = 1.03, *p* = 0.316, η_p_^2^ = 0.02). The prime × group interaction was not significant (*F*(4, 240) = 0.68, *p* = 0.607, η_p_^2^ = 0.01) and neither was the AOI × group interaction effect (*F*(1, 240) = 1.06, *p* = 0.307, η_p_^2^ = 0.02). The prime × AOI interaction effect was significant (*F*(3.46, 207.64) = 4.33, *p* = 0.004, η_p_^2^ = 0.07). The prime × AOI × group interaction was not significant (*F*(4, 240) = 1.45, *p* = 0.220, η_p_^2^ = 0.02). As residuals were not normally distributed, we conducted additional non-parametric (Mann–Whitney U) tests to examine group differences. The results confirmed that the groups did not differ on either of the parameters for “early dwell time” (all *p*s > 0.130; one-tailed). Single BH-corrected (*n* = 10) comparisons between prime conditions revealed that early dwell time on the eyes differed significantly between the happy and the angry prime condition (t(61) = −3.81, *p* < 0.001 (one-tailed), *d* = −0.48, CI [−0.75, −0.22]) and between the happy prime and the fearful prime condition (*z* = 2.57, *p* = 0.025 (one-tailed), *r* = 0.33). All other differences in early dwell time on the eyes were not significant (all BH-corrected (*n* = 4) *p*s ≥ 0.092; one-tailed). Early dwell time on the mouth differed significantly between the happy and the neutral prime condition (*t*(61) = 2.63, *p* = 0.015 (one-tailed), *d* = 0.33, CI [0.08, 0.59]) and the happy and the sad prime condition (*t*(61) = 3.17, *p* = 0.015 (one-tailed), *d* = 0.40, CI [0.14, 0.66]). All other comparisons were not significant (all BH-corrected (*n* = 10) *p*s ≥ 0.053; one-tailed). To account for the potentially confounding factor of the randomization sequence, we performed a non-parametric one-way ANOVA. The results indicated no effect of the randomization sequence (all *p*s > 0.062).

Analysis of “dwell time” data revealed no main effect of the prime (*F*(4, 240) = 0.367, *p* = 0.832, η_p_^2^ < 0.01), a significant main effect of the AOI (*F*(1, 240) = 98.01, *p* < 0.001, η_p_^2^ = 0.62), but no significant main effect of group (*F*(1, 60) = 1.40, *p* = 0.248, η_p_^2^ = 0.02). The prime × group interaction effect was not significant (*F*(4, 240) = 0.79, *p* = 0.533, η_p_^2^ = 0.01) and neither was the AOI × group interaction effect (*F*(1, 240) = 2.60, *p* = 0.112, η_p_^2^ = 0.04). Neither the prime × AOI × group interaction (*F*(4, 240) = 0.32, *p* = 0.865, η_p_^2^ < 0.01) nor the prime × AOI interaction were significant (*F*(4, 240) = 0.37, *p* = 0.830, η_p_^2^ < 0.01). The one-way ANOVA used to account for the potential confounding role of the randomization sequence indicated no effect of the randomization sequence on the “dwell time” data (all *p*s ≥ 0.657).

### 3.4. Correlation Analysis

A series of bivariate correlations was calculated to further analyze behavioral and eye-tracking data. As there was no effect of prime on either evaluative rating, “probability of first fixation,” or “dwell time,” the data were collapsed with respect to the primes for each parameter. All corresponding *p*-values are BH-corrected (*n* = 4). Bivariate correlations between evaluative ratings and the CTQ, BSL-23, BDI-II, and STAI-T revealed no significant correlations in the BPD group (all *p*s ≥ 0.368; two-tailed) or the NP group (all *p*s ≥ 0.052; two-tailed). No significant correlation was found between the “probability of first fixation” on the eyes or the mouth and the CTQ, BSL-23, BDI-II, and STAI-T in the BPD group (all *p*s ≥ 0.074; two-tailed) or the NP group (all *p*s ≥ 0.360; two-tailed). Interestingly, we found a significant correlation between “dwell time” on the mouth and the CTQ (*r* = 0.48, *p* = 0.024 (two-tailed), CI [0.15, 0.71]), the BSL-23 (*r* = 0.44, *p* = 0.026 (two-tailed), CI [0.10, 0.69]), and the STAI-T (*r* = 0.39, *p* = 0.043 (two-tailed), CI [0.04, 0.65]) in the BPD group. The correlation between “dwell time” on the mouth and the BDI-II was not significant in the BPD group (*r* = 0.08, *p* = 0.688 (two-tailed), CI [−0.29, 0.42]). Also, no significant correlations between clinical or affect measures and “dwell time” on the eyes were observed in the BPD group (all ps ≥ 0.985; two-tailed). Moreover, we found no significant correlations between “dwell time” on the eyes or the mouth and clinical or affect measures in the NP group (all *p*s ≥ 0.156; two-tailed).

Interestingly, the BSL-23 and the CTQ were not significantly correlated in either of the groups (all *p*s ≥ 0.296; two-tailed). Also, the STAI-T was not correlated with the CTQ or the BSL-23 in the BPD group (all *p*s ≥ 0.073). However, the STAI-T was correlated with the BSL-23 (*r* = 0.49, *p* = 0.010 (two-tailed), CI [0.16, 0.72]) but not with the CTQ (*r* = 0.19, *p* = 0.320 (two-tailed), CI [−0.19, 0.51]) in the NP group. Corresponding *p*-values are BH-corrected (*n* = 2).

In Table A5 in the Appendix A, we provide the correlations between “dwell time” on the eyes and the mouth and the CTQ, BDI, BSL-23, and the STAI-T calculated for each prime category separately. To examine the associations between eye-tracking data for the mouth and the eyes, we calculated the according bivariate correlations for each group separately. All following *p*-values are BH-corrected (*n* = 2). The results showed a significant negative correlation between “dwell time” on the eyes and the mouth for NPs (*r* = −0.49, *p* = 0.010 (two-tailed), CI [−0.72, −0.17]) but not for the BPD group (*r* = 0.020, *p* = 0.913 (two-tailed), CI [−0.34, 0.37]). “Probability of first fixation” on the eyes and the mouth was significantly correlated in the BPD group (*r* = −0.92, *p* < 0.001 (two-tailed), CI [−0.96, −0.83]) and the NP group (*r* = −0.99, *p* < 0.001 (two-tailed), CI [−0.99, −0.97]).

## 4. Discussion

This study aimed to investigate automatic emotional processing in individuals with BPD compared to NPs under conditions of unawareness of emotion stimuli. We applied an affective priming paradigm with the prime categories neutral, angry, fearful, sad, and happy faces, followed by neutral target faces. We analyzed evaluative and (early and late) gaze behavior concerning neutral target faces. Individuals with BPD evaluated target faces more negatively, irrespective of the preceding prime category compared to NPs. However, we found no effect of emotional prime presentation on evaluative ratings in either of the groups. Regarding early and late gaze behavior, we found no alterations in individuals with BPD compared to NPs. Independent of group, we found a priming effect in early gaze behavior (i.e., “early dwell time”).

Regarding the evaluative ratings of neutral target faces, we hypothesized that after being briefly presented with primes, individuals with BPD would evaluate neutral target faces more negatively when preceded by negative primes (i.e., angry, fearful, or sad faces) compared to NPs. As we observed no evidence for evaluative priming effects, we cannot draw conclusions about automatic emotional processing in BPD on the level of evaluative behavior. Findings from two studies by [29] ([29], [30]), who observed significant affective priming effects in their experiments, indicated no alterations in the automatic recognition and processing of emotional information in patients with BPD compared to NPs. In our study, neither group showed an effect of prime category on evaluative behavior. Thus, in our sample, prime stimuli were not efficient in affecting evaluative behavior. It must be noted that previous priming research with healthy and clinical samples has produced heterogeneous results concerning the effect of emotional facial expressions on evaluative behavior. Whereas there is evidence for a priming effect in some investigations (e.g., [58]; [70]), other studies did not find a priming effect on evaluative behavior (e.g., [11], [12]).

Our data suggests that, regardless of the preceding prime category, individuals with BPD evaluated neutral faces more negatively than NPs. This finding is in line with the negative interpretation bias frequently observed in individuals with BPD ([2]; [5]; [15]; [27]; [35]). Patients with BPD appear to be characterized by a general negative interpretation bias, perceiving neutral social information in a negatively biased manner. This could be due to negative cognitive schemas related to BPD ([3]; [72]) being activated in evaluative processes. Negative affect is a core symptom of BPD ([52]; [75]) and is associated with biases in interpretation processes ([10]). Interestingly, evaluative ratings were not associated with clinical or affect measures in either of the groups. This observation is in line with [30] ([30]) but only partially with [29] ([29]), who found positive associations of prime-based evaluative shifts with comorbid anxiety disorder. Against this background, the possible associations of automatic evaluative shifts with clinical symptoms and comorbidity in BPD remain to be clarified in further studies. It is noteworthy that NPs also exhibited, on average, a negative evaluation of neutral target faces. This finding is consistent with the existing literature on the interpretation of neutral facial expressions ([57]; [74]), indicating that neutral faces are not perceived as neutral but rather as negative.

The prime manipulation effectively altered early gaze behavior in both groups. Independent of group, participants dwelled longer on the eyes of neutral target faces after angry and fearful primes compared to happy primes during the first 500 ms. This replicates previous findings by our research team ([11], [12]). While both the eyes and the mouth are important for the recognition of fearful facial expressions ([32]; [77]), the mouth is considered the key region for identifying happy facial expressions ([77]). Therefore, our results are in line with the evidence on the diagnostically important regions for happy and fearful emotional facial expressions. Our data provides evidence that prime manipulation leads to increased early attention allocation towards the most important region for the recognition of anger (i.e., the eyes; [32]). Additionally, we observed increased early attention allocation towards the mouth of neutral target faces after happy primes compared to neutral and sad primes in both groups. This confirms the results of previous priming research ([11]) and is in line with the importance of the mouth for the recognition of happy facial expressions ([77]). Briefly presented emotional facial expressions—both negative and positive—appear to activate an automatic mechanism that is attuned to the distribution of diagnostic facial features, thereby guiding early attentional orientation towards these regions (see also [12]). Contrary to our hypothesis, however, we observed no significant differences in early gaze orientation towards the eyes of target faces when preceded by a neutral prime versus a fearful, angry, or sad prime. Similarly, there were no differences in gaze orientation towards the eyes of target faces when primed with a happy versus a sad face. Nevertheless, our data is consistent with an early, expression-specific attentional bias toward facial regions that are most relevant for identifying the corresponding emotional expression. It is interesting to note that the amygdala appears to play a crucial role in reflexive gaze initiation and reflexive orienting towards facial features, especially towards fearfully widened eyes ([37]; [38]). Specific regions within the amygdala seem to be specialized in actively making eye contact ([20]). Although BPD patients manifest, in general, stronger amygdala activation during emotion processing compared to NPs (see, for example, [25], for a recent meta-analysis), in our study, we observed no increased attention allocation to the eyes of fearful or angry faces in BPD patients in comparison with NPs. The question arises as to whether this finding could indicate that BPD patients do not respond with increased amygdala activation when emotional stimuli are of low intensity or not consciously perceived.

Contrary to our hypothesis, we found no group differences between individuals with BPD and NPs in early gaze behavior toward the eyes or mouth. Note that the corresponding effect was far from significant in “probability of first fixation.” Based on our findings, individuals with BPD exhibit a level of vigilance toward briefly presented masked facial expressions comparable to that of NPs. Although emotional hypersensitivity in BPD ([60]) may contribute to altered early gaze patterns to emotional primes, our findings do not support the notion of increased sensitivity in automatic emotional processing. Moreover, individuals with BPD did not look with a higher probability at the eyes first, contradicting the results of [79] ([79]) and, in part, the findings of the studies by [8] ([8], [9]). On the other hand, our findings are in line with the view that early gaze behavior (as assessed by the “probability of first fixation” and “early dwell time”) concerning central facial features is not altered in BPD (cf. [92], for free viewing with multiple faces; [14], for socioemotional content). As the eyes can be interpreted as the most diagnostically important feature for recognizing negative emotions ([32]; [77]) and hence social threat, our early gaze behavior data do not support the threat-hypersensitivity hypothesis ([9]) and that of early vigilance for social threat ([79]) in BPD. Our results are in line with the view that automatic emotional processing is not altered in BPD (see also [71]). In our study, we found no associations of early gaze behavior (i.e., “probability of first fixation”) towards the eyes or the mouth with clinical or affect measures in either of the groups.

Regarding late attention allocation (i.e., “dwell time”), contrary to our hypothesis, we found no evidence that individuals with BPD dwell longer on the eye region compared to NPs. At the level of late attention deployment, our results are in line with those of [79] ([79]), which also indicate no difference between individuals with BPD and healthy controls in late attention deployment towards the eyes of neutral and emotional facial expressions. Thus, we cannot confirm the assumption that individuals with BPD manifest increased attention maintenance on the eye region, which might be part of a general vigilance for social threat or an emotional hypersensitivity ([60]).

Taken together, our evaluative behavior data shows, as expected, a more negatively biased perception of facial expressions in BPD compared with NPs. Our results concerning early gaze behavior do not indicate heightened vigilance toward the eye region in BPD when looking at and evaluating neutral faces compared to NPs. This finding seems to contrast with results from previous eye-tracking research indicating that BPD patients direct their gaze more quickly to the eyes of very briefly presented neutral faces ([9]; [79]). The diverging results could be due to differences in experimental procedures. In the studies of Bertsch et al. and Seitz et al., participants had to identify the emotional quality of facial expressions (angry, fearful, happy, or neutral), and face presentation durations were 150 ms or 5000 ms. In our study, participants had to evaluate the valence of faces, which were displayed for 1850 ms. It is possible that when more complex decision processes are required and facial stimuli are shown very briefly (for 150 ms), BPD patients, compared to NPs, direct their attention more quickly to the eye region, which is the most informative facial feature concerning emotion recognition ([16]; [41]). In contrast, if the task is less complex and more time is available to complete it (as in our experiment), then BPD patients may not manifest faster orienting to the eyes compared to NPs.

In the BPD group, traumatic experiences in childhood, trait anxiety, and severity of borderline symptoms were positively associated with attention allocation to the mouth. This increased attention to a region of the face that, unlike the eyes, is not of crucial importance for establishing interpersonal contact ([18]; [45]) and emotion recognition ([16]; [41]) could represent a protective mechanism in patients with severe borderline psychopathology or high dispositional anxiety that helps to reduce social contact and the perception of potential threats. When looking at faces, people clearly tend to focus more on the eyes than the mouth or other facial features ([46]; [89]). Interestingly, experiences of childhood maltreatment have been found to be linked to less gaze fixation on the eye area ([87]). Note that BSL-23, STAI-T, and CTQ were not correlated in the BPD group. Therefore, trait anxiety, childhood trauma, and severity of borderline symptoms manifest rather independent associations with attention allocation towards the mouth, a face area of secondary importance for recognizing emotion and regulating social interaction. The correlation between borderline symptom severity and attention to the mouth observed in our study could be part of a late threat avoidance strategy, which has previously been found in individuals with BPD ([9]; [15]).

This study has several limitations. First, although no group differences in early or late gaze behavior were observed, the existence of a corresponding effect cannot be ruled out. Since the observed effect was small and did not reach statistical significance, larger sample sizes might be needed to uncover subtle interaction effects. It is also possible that the absence of group differences in priming effects in our study might be attributable to insufficient task sensitivity, specific characteristics of the target faces, the composition of the stimulus set, or the partially conscious processing of emotional stimuli. Second, antidepressant medication in the BPD group affected gaze behavior, confounding our results. Antidepressant medication intake is known to facilitate processing of positive emotional information (e.g., [91]) and therefore may decrease priming effects, particularly when negative emotional content is used as a prime. Third, we did not investigate whether comorbid mental disorders have an impact on our results. Future studies with larger samples are necessary to form subgroups of patients and to examine the effect of comorbid mental disorders (e.g., comparing individuals with BPD and major depression with individuals with BPD and anxiety disorders and individuals with BPD without comorbid mental disorders) and antidepressant medication systematically. Fourth, in our study, awareness of emotional prime faces was assessed only via an interview (measuring self-reported unawareness), i.e., not via an objective prime detection task. Although prior research indicates subliminal prime processing for masked prime stimuli (see [50]), we cannot rule out the possibility that some participants were aware of the primes. Fifth, in our investigation, we did not compare gaze and evaluative behavior between BPD patients and NPs as a function of biological sex since the number of male patients included was low (5 out of 31). Such comparisons could be valuable in future studies as women have shown stronger evaluative shifts in response to happy faces in affective priming tasks ([31]) and have been shown to exhibit enhanced neural responses to neutral faces primed by subliminally presented fearful faces ([88]). However, the results of our previous eye-tracking study based on the affective priming task ([11]) indicated no effects of biological sex on prime-valence dependent early or late gaze behavior in NPs. Additionally, our investigation is limited to neutral target faces. Future studies could administer emotional target faces in masked emotional priming research with BPD patients. Finally, we want to point out that future eye-tracking research on emotion recognition from faces and gaze behavior in BPD patients could implement an ideal observer analysis in face categorization tasks ([19]). Such an approach allows for strong conclusions to be drawn about whether the gaze strategy adopted by BPD patients is driven by diagnostically relevant features or by other factors such as tendencies to avoid threatening information in the eyes.

## 5. Conclusions

The present eye-tracking study aimed to investigate automatic emotional processing in BPD under conditions of unawareness of emotion stimuli using an affective priming paradigm based on facial expressions. Our data indicates a priming effect in early gaze behavior such that automatic attentional orientation was directed toward the most diagnostically relevant region of the facial expression corresponding to the preceding prime when viewing the subsequently presented neutral target face. However, our results indicate that individuals with BPD do not show alterations in automatic attention orienting due to covert facial emotions. Independent of prime presentation, our results demonstrate that individuals with BPD are characterized by an increased negative interpretation of neutral faces compared to NPs. The present eye-tracking data does not support the assumption of an early or late hypervigilance towards the eyes in BPD. Our study provides initial evidence that in patients with BPD, severity of borderline psychopathology, trait anxiety, and childhood maltreatment could be associated with increased attention allocation towards the mouth.

## Figures and Tables

**Figure 1 behavsci-15-01268-f001:**
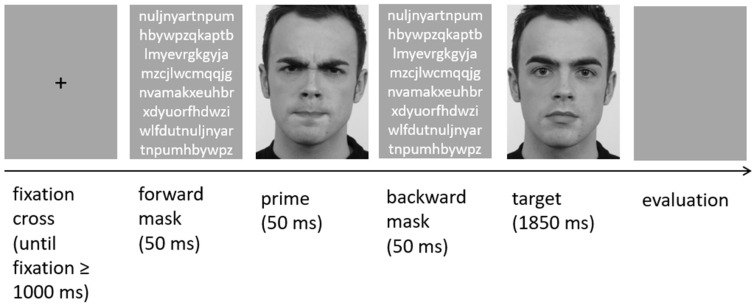
Schematic illustration of a single trial in the affective priming experiment. The face images were taken from the Radboud Faces database (RaFD; [54]). In scientific publications, RaFD images can be presented as stimulus examples.

**Figure 2 behavsci-15-01268-f002:**
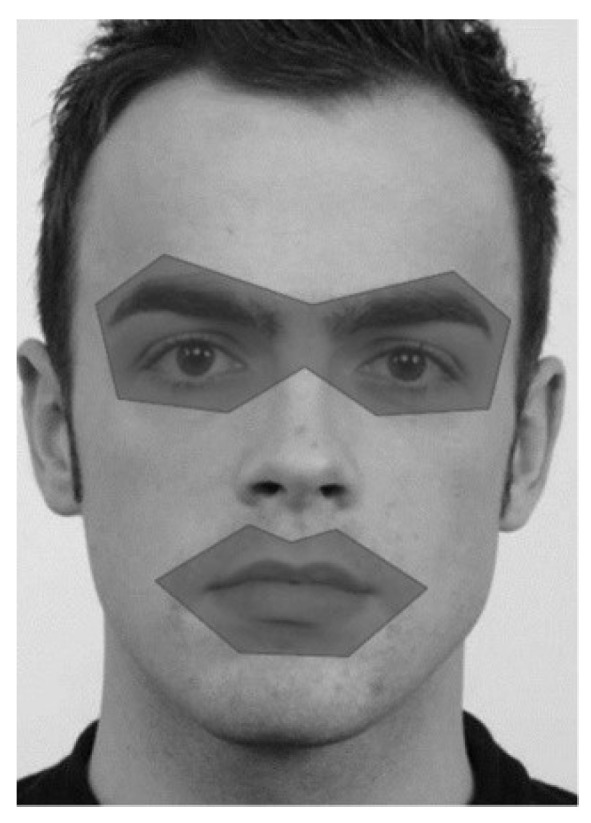
Illustration of the AOIs for the eyes and the mouth. The face image was taken from the Radboud Faces database (RaFD; [54]).

**Figure 3 behavsci-15-01268-f003:**
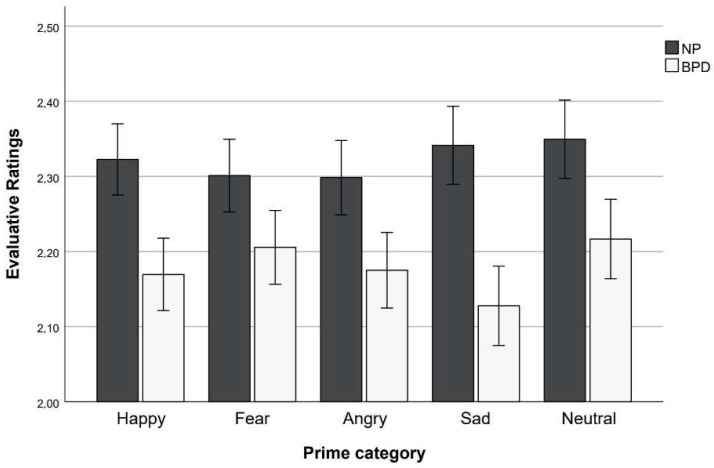
Evaluative ratings for prime categories and study groups. Error bars represent standard error.

**Figure 4 behavsci-15-01268-f004:**
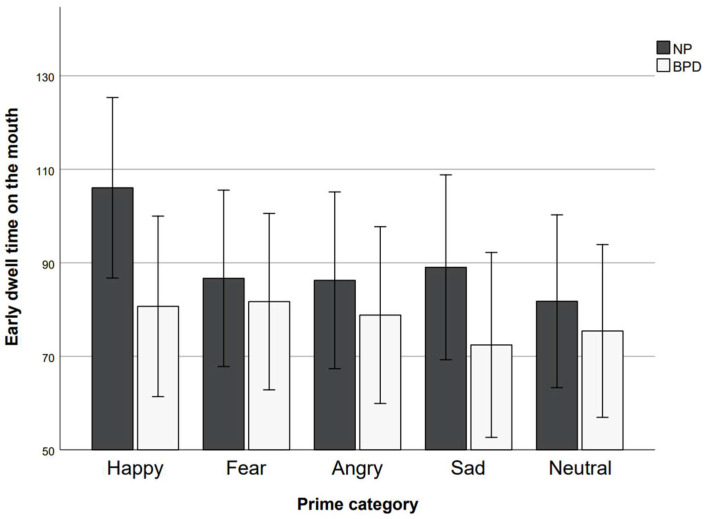
Early dwell time on the mouth in ms for prime categories and study groups. Error bars represent standard error.

**Figure 5 behavsci-15-01268-f005:**
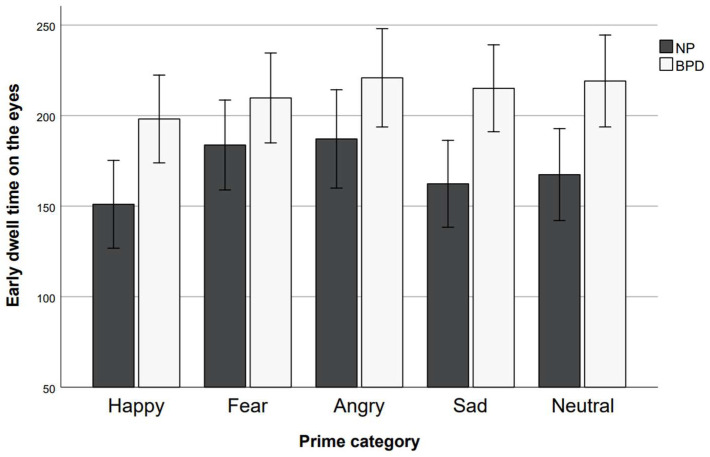
Early dwell time on the eyes in ms for prime categories and study groups. Error bars represent standard error.

**Table 1 behavsci-15-01268-t001:** Demographic and psychological characteristics of the study groups (mean with standard deviation and range).

	BPD (N = 31)			NP (N = 31)				
	M	SD	Range	M	SD	Range	*d*	95% CI
Age	27.81	6.09	18–44	26.65	5.16	20–42	−0.21	[−0.70, 2.94]
School years	11.71	1.35	9–15	12.23	0.81	9–13	0.47	[−0.04, 0.97]
TMT-B	67.74	24.68	35–132	60.32	19.71	34–121	−0.33	[−0.83, 0.17]
BDI-II	21.29	8.07	2–35	3.45	2.74	0–9	−2.96 *	[−3.68, −2.23]
STAI-T	60.32	8.34	41–77	34.68	6.92	25–57	−3.45 *	[−4.12, −2.56]
BSL-23	1.77	0.75	0.48–3.17	0.17	0.14	0–0.52	−2.99 *	[−3.71, −2.25]
CTQ	63.16	18.09	35–113	33.42	7.59	25–58	−2.14 *	[−2.77, −1.51]

* Significant group differences at *p* < 0.01; TMT-B = Trail Making Test Part B; BDI-II = Beck Depression Inventory-II; STAI-T = State–Trait Anxiety Inventory, trait version; BSL-23 = Borderline Symptom List; CTQ = Childhood Trauma Questionnaire.

**Table 2 behavsci-15-01268-t002:** Comorbid axis I and axis II disorders and antidepressant medication intake in the BPD group.

**Comorbid Axis-I Disorder**	** *n* **
Affective disorder	17
Anxiety disorder	14
PTSD	9
Eating disorder	12
Somatoform disorder	5
**Comorbid personality disorder**	** *n* **
Obsessive–compulsive	12
Depressive	14
Self-insecure	11
Dependent	4
Antisocial	4
Paranoid	3
Narcissistic	4
Schizotypal	3
Passive–aggressive	5
Histrionic	1
**Medication intake**	** *n* **
SSRI	9
SNRI	2
SDRI	1
TeCA	1
TCA	1

SSRI = selective serotonin reuptake inhibitor; SNRI = serotonin–noradrenaline reuptake-inhibitor; SDRI = serotonin–dopamine reuptake inhibitor; TeCA = tetracyclic antidepressant; TCA = tricyclic antidepressant.

**Table 3 behavsci-15-01268-t003:** Descriptive statistics of “probability of first fixation” and in the affective priming experiment for study groups as a function of the prime category and area of interest.

	BPD (N = 31)			NP (N = 31)		
Prime Category–AOI	M	SD	Range	M	SD	Range
Happy–mouth	0.29	0.32	0–1	0.37	0.35	0–1
Happy–eyes	0.69	0.31	0–1	0.62	0.35	0–1
Fearful–mouth	0.27	0.33	0–1	0.33	0.32	0–0.83
Fearful–eyes	0.70	0.33	0–1	0.66	0.33	0.17–1
Angry–mouth	0.27	0.31	0–1	0.34	0.34	0–1
Angry–eyes	0.69	0.31	0–1	0.64	0.35	0–1
Sad–mouth	0.25	0.32	0–1	0.35	0.33	0–1
Sad–eyes	0.72	0.31	0–1	0.63	0.33	0–1
Neutral–mouth	0.26	0.29	0–0.92	0.33	0.34	0–1
Neutral–eyes	0.70	0.30	0–1	0.66	0.34	0–1

AOI = area of interest.

**Table 4 behavsci-15-01268-t004:** Descriptive statistics of “dwell time” (ms) in the affective priming experiment for study groups as a function of the prime category and area of interest (mean with standard deviation).

	BPD (N = 31)			NP (N = 31)		
Prime Category–AOI	M	SD	Range	M	SD	Range
Happy–mouth	255.2	167.2	24–591.1	289.9	220.0	0–846.6
Happy–eyes	844.8	290.6	0–1298.2	697.1	320.5	182.0–1365.4
Fearful–mouth	258.7	167.0	0–666.6	284.9	190.7	32.7–784.0
Fearful–eyes	842.8	325.9	0–1338.5	725.7	337.4	144.3–1220.5
Angry–mouth	248.9	160.4	0–603.9	291.5	210.0	13.7–799.2
Angry–eyes	839.6	313.7	0–1395.8	720.3	351.4	143.8–1342.4
Sad–mouth	257.0	170.2	0–563.3	289.0	194.2	19.7–805.9
Sad–eyes	839.7	301.1	24.7–1327.5	695.8	340.3	117.0–1333.1
Neutral–mouth	249.2	152.0	18.0–551.6	292.8	223.5	0–950.8
Neutral–eyes	840.4	299.1	16.7–1275.1	724.3	314.1	187.3–1229.6

AOI = area of interest.

## Data Availability

The datasets used and/or analyzed during the current study are available from the corresponding author on reasonable request.

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
