# Peer review of "Gaze and Evaluative Behavior of Patients with Borderline Personality Disorder in an Affective Priming Task"

_behavsci, 2025, doi:10.3390/bs15091268_

Round 1
Reviewer 1 Report (Previous Reviewer 1)
Comments and Suggestions for Authors
Introduction
The introduction first provides the relevance of emotion processing impairment in individuals with Borderline Personality Disorder (BPD) to lay the groundwork for the use of masked affective priming and eye-tracking tools to explore unconscious prejudice within social perception.
- The conduct of current may be made easier by explicitly making the connection between the empirical approach and theoretical framework. More justification would be needed for masking primes and AOI-based gaze measures.
- The gap in research is neither defined nor cogently defined as positive. I suggest including a bridging paragraph that rounds up existing work and demonstrates how this research extends or diverges from existing work.
Literature Review
The article blends background information regarding BPD, emotion processing, and eye-tracking but lacks a clear literature review section. The review is embedded in the Introduction.
- The integration between domains is weak. The review separates affective priming, gaze behavior, and BPD into distinct strands.
- The scope of the literature is relatively limited, specifically in relation to masked priming in clinical populations.
- Missing or unused citations in masked priming experiments
Methodology
The method section defines participant details, stimuli, apparatus, and procedure for the rating judgment and eye-tracking task.
- Trial timing and trial structure: ISI, number of trials per condition, inter-trial intervals, and fixation cross duration is scattered. Include a representative visual timeline per trial (e.g., prime → mask → target → rating).
- Masking verification: The experiment exclusively relies on self-reported unawareness, without any forced-choice detection test.
- State the limitation explicitly; quote literature that supports subliminal perception.
- AOI definition: Failed to adequately describe how AOIs were defined or normalized. Define manual or software-defined; quote inter-coder reliability where necessary.
- Calibration: Failure to state calibration quality or standards. Describe the eye-tracking calibration procedure briefly, including drift correction.
- Sample power: Post hoc analysis is given, but preferably would be a priori.
- Reframe the power statement as exploratory, or provide a retrospective justification based on prior work effect sizes.
Authors carried out mixed-design ANOVAs and Pearson correlations to examine similarities and differences in the effect of emotional primes and between-group differences on gaze and evaluative behavior.
- ANVOA assumption checks (normality, sphericity, homogeneity) are not provided.
- Run Mauchly's test of sphericity and use Greenhouse-Geisser corrections if significant. Residual normality and homogeneity; Levene's test and QQ plots.
- There are no effect sizes and confidence intervals. Report partial η² or Cohen's d with 95% CIs for all main effects and interactions.
- The assumptions of normality, linearity of correlation remain untested. Check assumptions or use Spearman's rho as appropriate.
- Multiple comparison corrections are not provided. Use Holm–Bonferroni or FDR control in correlation tables to limit Type I error.
Results
Results are distinctly separated among evaluative ratings, gaze behavior, and correlations with clinical characteristics.
- AOI measure descriptive statistics by condition/group are not distinctly tabulated. Summarize tables of means, SDs, and EMMs by conditions.
- Null effects are at times underinterpreted. Describe potential reasons for not observing group × prime effects (e.g., task sensitivity, masking effectiveness).
- Figures could be improved with increased labeling or visual group/condition contrast.
Discussion
- Discussion concludes primary findings, mentions a lack of group-specific priming effects, and describes interpretation bias and clinical correlations.
- Shift from findings to implications is discontinuous.
- Open with a brief summary paragraph of primary findings.
- Theoretical underpinnings for null findings are weak.
- Describe possible cognitive models (e.g., top-down control, attentional biases) to account for findings.
Limitations section is concise and disjointed. Include a clear "Limitations and Future Directions" paragraph covering masking awareness, sample size, comorbidities, and absence of emotional target faces.
Author Response
Introduction
The introduction first provides the relevance of emotion processing impairment in individuals with Borderline Personality Disorder (BPD) to lay the groundwork for the use of masked affective priming and eye-tracking tools to explore unconscious prejudice within social perception.
The conduct of current may be made easier by explicitly making the connection between the empirical approach and theoretical framework. More justification would be needed for masking primes and AOI-based gaze measures.
Our RESPONSE: Our previous research (Bodenschatz et al., 2019, 2020) has shown that our specific masking procedure combined with our selection of two Areas of interest (eyes and mouth) worked successfully to measure changes in gaze behavior depending on emotion prime category in a theoretically meaningful way. In our manuscript, we mention that the same experimental procedure has been applied by us in previous investigations (see l.127-130, l. 219-221, l. 391-393). Past research has demonstrated that the eyes and the mouth are central features in the recognition of facial emotions and are viewed preferentially (Buchan et al., 2007; Hall et al., 2010; Scheller et al., 2012). Please also note that reviewer #3 has criticized our introduction for being too long and extensive and suggested cuts.
The gap in research is neither defined nor cogently defined as positive. I suggest including a bridging paragraph that rounds up existing work and demonstrates how this research extends or diverges from existing work.
Our RESPONSE: Following the reviewer’s suggestion, we address the fact that there is a gap in previous research and clarify in our revised manuscript what distinguishes our study from previous research in the field (see l.196-201).
Literature Review
The article blends background information regarding BPD, emotion processing, and eye-tracking but lacks a clear literature review section. The review is embedded in the Introduction.
Our RESPONSE: From our point of view, it is not unusual for a research paper not to have a separate literature review section in the introduction, but to present previous findings on the various topics of the paper consecutively in the introduction.
The integration between domains is weak. The review separates affective priming, gaze behavior, and BPD into distinct strands.
Our RESPONSE: We agree with the reviewer that a stronger linking of the paragraphs and topics could further improve our introduction, but it would take up additional space and conflict with wishes of reviewer #3 to shorten the introduction. In the penultimate paragraph of our introduction section (l.203-233), in which we specify our research questions and hypotheses, we summarize and integrate the relevant results of previous research.
The scope of the literature is relatively limited, specifically in relation to masked priming in clinical populations.
Our RESPONSE: In our opinion, our manuscript already presents a number of relevant studies on clinical populations that employed the affective priming paradigm (see l.74-85). A comprehensive overview, which includes all existing research on the topic, seems to be beyond the scope of our experimental research paper that describes data of one experiment and examines behavior of one clinical population (i.e., BPD patients).
Missing or unused citations in masked priming experiments
Our RESPONSE: We are not sure which other literature on masked priming should be cited here. If the studies the reviewer has in mind are essential, we kindly ask the reviewer to explicitly name these studies (with author and publication year).
Methodology
The method section defines participant details, stimuli, apparatus, and procedure for the rating judgment and eye-tracking task.
Trial timing and trial structure: ISI, number of trials per condition, inter-trial intervals, and fixation cross duration is scattered. Include a representative visual timeline per trial (e.g., prime → mask → target → rating).
Our RESPONSE: Please note that the fixation cross duration was not fixed. Also, the duration of the evaluation phase was not fixed. Therefore, description of ISI and ITI are not possible by providing fixed or “exact” numbers. Note that we do have a visual timeline as the reviewer suggested (see Figure 1). We now edited Figure 1 to make the structure of the trials clearer by adding the information that the fixation cross was shown until a fixation of 1000ms. We additionally edited section 2.3 to make the description of the trial structure clearer (see l.328-341).
Masking verification: The experiment exclusively relies on self-reported unawareness, without any forced-choice detection test.
State the limitation explicitly; quote literature that supports subliminal perception.
Our RESPONSE: We included reliance on self-report when assessing prime unawareness as limitation of our study in our revised discussion section (see l.758-760) and added literature that supports subliminal processing of masked primes (see l.388-390 and l.760-762).
AOI definition: Failed to adequately describe how AOIs were defined or normalized. Define manual or software-defined; quote inter-coder reliability where necessary.
Our RESPONSE: We revised the information given in our methods section on how AOIs were defined. Specifically, we included the information that AOIs were drawn manually by a single coder and that the procedure was standardized (see l.415-420) Please note that since there was only one coder in our study, we are not able to provide information on inter-coder reliability.
Calibration: Failure to state calibration quality or standards. Describe the eye-tracking calibration procedure briefly, including drift correction.
Our RESPONSE: Please note that we excluded 8 participants who exceeded a calibration deviation of 1.0° (see l.280-281). Responding to the reviewer’s comment, we revised the description of the calibration and validation procedure to make it clearer – including also information on drift correction (see l.375-380).
Sample power: Post hoc analysis is given, but preferably would be a priori. Reframe the power statement as exploratory, or provide a retrospective justification based on prior work effect sizes.
Our RESPONSE: Please note that we made an a-priori power analysis for the within-group priming effect based on the effect size reported by Bodenschatz et al. (2020). In our revised manuscript, we reframed the statistical power statement as “exploratory” (see l.264).
Authors carried out mixed-design ANOVAs and Pearson correlations to examine similarities and differences in the effect of emotional primes and between-group differences on gaze and evaluative behavior.
ANVOA assumption checks (normality, sphericity, homogeneity) are not provided.
Our RESPONSE: Please see Table A6 for assumption tests. There, we report all assumption tests conducted and corresponding violations in a summarized but informative manner. Please note that reporting statistical details of single assumption tests would be very space consuming and is not common practice in research articles in the field.
Run Mauchly's test of sphericity and use Greenhouse-Geisser corrections if significant. Residual normality and homogeneity; Levene's test and QQ plots.
Our RESPONSE: We did run Mauchly´s test for sphericity and used GG-correction if significant. Results of the Mauchly´s test can be found in Table A6 as well. Information on GG-correction can be found in l.455-456, GG correction was administered in several cases: see l.510, 512, 515 and 528. In Table 6, we also provide information on the Levene´s test and residual normality. We refrain from reporting results of QQ-Plot analysis because a) it is not common practice of research articles in the field; b) in our case it provides no further information when added to the normality tests; and c) it would need much additional space.
There are no effect sizes and confidence intervals. Report partial η² or Cohen's d with 95% CIs for all main effects and interactions.
Our RESPONSE: We agree with the reviewer that we did not report 95% CIs for the ANOVA main effects and interactions. Please note that in SPSS there is no output option for 95% CI concerning ANOVA main effects or interactions. However, we do not see it as necessary to add these 95% CIs when already providing p-values and effect sizes and reporting 95% CIs for post-hoc tests.
The assumptions of normality, linearity of correlation remain untested. Check assumptions or use Spearman's rho as appropriate.
Our RESPONSE: Please check Table A6 and section 2.7. We did report normality tests. Linearity was tested by scatterplots but is not provided in the manuscript because it would require too much space. As we mention in l.467-468 of our statistical analysis section, we used Spearman´s rho in case of assumption violations.
Multiple comparison corrections are not provided. Use Holm–Bonferroni or FDR control in correlation tables to limit Type I error.
Our RESPONSE: As pointed out in l.469-470 of our statistical analysis section, we did apply correction of p-levels in all cases of multiple testing (see also l. 532, 537, 541, 567, 586, and 592). We used Benjamini-Hochberg correction, which is a legitimate method of reducing type I error.
Results
Results are distinctly separated among evaluative ratings, gaze behavior, and correlations with clinical characteristics.
AOI measure descriptive statistics by condition/group are not distinctly tabulated. Summarize tables of means, SDs, and EMMs by conditions.
Our RESPONSE: As summarizing tables would result in very large tables and therefore decrease clarity, we refrain from summarizing the tables. However, to make descriptive statistics presentation clearer, we have changed the layout of the corresponding tables (see Tables 3 and 4).
Null effects are at times underinterpreted. Describe potential reasons for not observing group × prime effects (e.g., task sensitivity, masking effectiveness).
Our RESPONSE: We thank the reviewer for the feedback. We acknowledge that discussion of potential reasons for not observing group x prime interaction effects was missing. We now added additional potential reasons for not observing group differences in our limitation section (see l.746-749).
Figures could be improved with increased labeling or visual group/condition contrast.
Our RESPONSE: We revised Figure 1 for better clarity but left the remaining figures as they were since we believe that they are already clear.
Discussion
Discussion concludes primary findings, mentions a lack of group-specific priming effects, and describes interpretation bias and clinical correlations.
Shift from findings to implications is discontinuous.
Open with a brief summary paragraph of primary findings
.
Our RESPONSE: Following the reviewer’s suggestion, we changed the first paragraph of our discussion and now briefly present all main findings at the beginning of the discussion section (see l. 606-611).
Theoretical underpinnings for null findings are weak.
Our RESPONSE: In our view, null findings should not be seen as evidence for the absence of an effect, at least in traditional inference statistics (e.g., Harms & Lakens, 2018; J Clin Transl Res). Against this background, null effects should be interpreted parsimoniously and cautiously. Moreover, since reviewer #3 suggested shortening our discussion, we do not want to extend the discussion section with further potential meanings of null findings.
Describe possible cognitive models (e.g., top-down control, attentional biases) to account for findings.
Our RESPONSE: Unfortunately, the reviewer remains unclear here, what he/she means by “findings”. In response to the comment of the reviewer, we added a neurocognitive interpretation of our main finding concerning gaze behavior to our revised discussion section (see l.669-679).
Limitations section is concise and disjointed. Include a clear "Limitations and Future Directions" paragraph covering masking awareness, sample size, comorbidities, and absence of emotional target faces.
Our RESPONSE: Following the reviewer’s suggestion, we restructured and expanded the limitation section at the end of our discussion (see l.743-773). We now include self-reported prime unawareness and use of neutral target faces as potential limitations of our study (see l.758-762 and l.771-773) – in addition to sample size and comorbidities that also represent limitations.
In response to the reviewer’s suggestion, our manuscript underwent professional editing by a specialized MDPI language service to improve our English.
We thank the reviewer for the large number of valuable comments and suggestions.
Reviewer 2 Report (Previous Reviewer 3)
Comments and Suggestions for Authors
I commend the authors for their efforts to implement various suggested analyses and address issues where appropriate. I am satisfied with the changes in the manuscript and believe this research will be an important contribution to the area of gaze behaviors in BPD and emotional face perception.
Author Response
I commend the authors for their efforts to implement various suggested analyses and address issues where appropriate. I am satisfied with the changes in the manuscript and believe this research will be an important contribution to the area of gaze behaviors in BPD and emotional face perception.
Our RESPONSE: We thank the reviewer for the positive feedback.
Many thanks to the reviewer for sharing his/her experience with us.
Reviewer 3 Report (Previous Reviewer 2)
Comments and Suggestions for Authors
Dear authors,
The overall structure of the manuscript is sound, and the content is well developed. That said, some paragraphs in the Introduction and Discussion are relatively long. Breaking them into shorter, more focused sections would improve clarity, support pacing, and make the text more accessible to readers. Otherwise, manuscript is ready for publication.
Great work!
Author Response
Dear authors,
The overall structure of the manuscript is sound, and the content is well developed. That said, some paragraphs in the Introduction and Discussion are relatively long. Breaking them into shorter, more focused sections would improve clarity, support pacing, and make the text more accessible to readers. Otherwise, manuscript is ready for publication.
Great work!
Our RESPONSE: We understand that the reviewer suggests shortening some parts of the introduction and discussion to make our paper more focused and concise. Unfortunately, reviewer #1 requests further extensions of the introduction and the discussion, so that we must refrain from shortening these sections.
In sum, we thank the reviewer for his/her important methodological suggestions during the review process that improved our manuscript.
Round 2
Reviewer 1 Report (Previous Reviewer 1)
Comments and Suggestions for Authors
The authors have more than adequately addressed my comments and concerns.
This manuscript is a resubmission of an earlier submission. The following is a list of the peer review reports and author responses from that submission.
Round 1
Reviewer 1 Report
Comments and Suggestions for Authors
Introduction
The introduction presents the central constructs of borderline personality disorder (BPD), affective priming, and gaze behavior. It highlights the significance of generating understanding of attentional and affective processes in BPD and stipulates the aim of the study to examine gaze and affect ratings following subliminal emotional primes.
- The theoretical framework is not integrated across measures. BPD, gaze behavior, and affective priming are reported together but not as components of a single theoretical rationale.
- The affective priming task has poor theoretical rationale. The authors need to provide more references to justify why they use masked affective priming in clinical samples.
- The introduction ends without clearly formulated research questions or hypotheses, so it is not possible to evaluate how well theory and design are matched.
Literature Review
The literature review superficially includes emotional dysregulation in BPD, affective interpretation biases, and evidence of clinical populations' eye gaze.
- The breadth of literature is limited. The eye-tracking studies at the large scale in BPD or comparable populations are not addressed. Extend the review to eye-tracking studies with mixed results in BPD
- The review does not have contradiction of prior research of gaze behavior (e.g., hypervigilance vs. avoidance of the eyes). Enlarge the review to incorporate eye-tracking research that produces inconsistent results in BPD
- No reference to prior research into affective priming in BPD or other psychopathologies that eliminates the function of the task. Conclude with a synthesis paragraph connecting reviewed literature to study goals.
Methodology
The research applies a mixed design involving masked affective priming and an eye-tracking paradigm. The sample involves 31 BPD participants and 31 education- and age-matched controls. Eye patterns and ratings of affect were obtained after subliminal presentation of facial expressions of emotion.
- Stimulus presentation procedure (prime, 50 ms, masking) is inadequate. There is no figure or trial sequence schematic. Add a trial timeline diagram and a definition of masking parameters.
- AOIs (regions of interest) in eye-tracking are poorly defined, which restricts reproducibility. Elaborate on AOI construction process and save fixation detection thresholds.
- There is no mention of inclusion/exclusion criteria, determination of sample size, or power analysis. Perform and report a power analysis to determine the sample size.
- Medication and comorbidity in the BPD group are noted but not addressed analytically. Add sensitivity analyses or at least a discussion of how medication and comorbidities can confound results.
- Assumption tests (homogeneity, normality, sphericity) are not reported. Report assumption tests (Shapiro–Wilk, e.g.).
- No effect sizes or confidence intervals are provided, which restricts interpretability. Report partial η², Cohen's d, r², and 95% CIs throughout.
- Some of the correlation between emotion conditions and comparisons are not corrected. Apply FDR correction or Bonferroni correction where necessary.
Discussion
The debate covers the absence of priming effects, increased dwell on the mouth in BPD, and trauma and anxiety associations. It places gaze behavior within attentional avoidance and interpersonal hypersensitivity.
- Overextends mouth fixation interpretations to "avoidance" with no experimental evidence. Redo conclusions in a more tentative way (e.g., "may suggest avoidance").
- Does not include outcomes within the dominant theoretical models of BPD (e.g., biosocial theory, threat detection models).
- Underestimates the significance of null results for affective processing models of BPD. Consider how null results undermine or clarify prevailing conceptions of automatic affective processing in BPD.
Reviewer 2 Report
Comments and Suggestions for Authors
This a well-conceived and carefully executed study exploring automatic emotion processing in individuals with borderline personality disorder (BPD), using an affective priming paradigm and eye-tracking measures. The manuscript is technically sound, comprehensive, and written with clarity. The theoretical framing is strong, and with good methodological rigor.
General Impression:
The manuscript is ready as it is. I only offer a few minor suggestions below to enhance clarity and flow for readers less familiar with the technical aspects.
See PDF

Reviewer 3 Report
Comments and Suggestions for Authors
This manuscript investigates automatic emotion processing in borderline personality disorder (BPD) using eye tracking with an affective priming paradigm. The research question is well-conceived, and the general research direction has the potential to inform our understanding of the role of attention in emotion perception in BPD. However, the authors used an experimental manipulation that did not work, limiting what we can learn from this data. Further, I worry that the gaze analysis was not sensitive enough to capture fine-grained changes that could be relevant. I have detailed these issues below.
Major Issues:
- Failure of priming manipulation: In this task, both BPD and healthy observers did not demonstrate variations in the ratings for faces due to the prime. This puts the reader in a difficult situation - did the prime have any impact on the observers at all? Further, since there is literature supporting that priming does impact face emotion perception, there is a strong expectation to see such an effect. Therefore, there may have been subtle or transient effects that the experimental design failed to capture. While the authors describe possible reasons for not seeing the priming effect, we need to interpret this null result with caution. The absence of a significant effect does not imply a lack of emotional modulation. The interpretation of any eye-tracking effects becomes challenging: is the observed pattern of results due to or despite these subthreshold (did not reach significance) emotional changes?
- Insufficient spatial resolution in Eye tracking analysis: The current analysis focuses primarily on aggregate gaze duration within coarse areas of interest (AOIs), namely the eyes and mouth regions of the neutral target face. This binary AOI approach may obscure more subtle patterns of gaze modulation. For example, gaze shifts could be much more subtle, with the first fixation shifting to the nose tip instead of the mouth. Prior research actually shows that even for emotions primarily conveyed by the mouth region, human first fixations tend to land near the nose tip (covering the mouth with peripheral vision) to optimize emotion recognition (see Peterson and Eckstein 2012; PNAS). Thus, a finer-grained analysis of the first fixation positions may be better suited. In this context, I was also not sure about the size of faces on the observer's eyes (the paper provides the size of the faces on the screen but the distance of the observer from the screen is not given). If the faces subtended a small visual angle on the observer's eye, it is possible that the observer did not need to make an overt saccade to the mouth to access the mouth information.
- No comparison of gaze with the true distribution of emotion-relevant information. To draw definitive conclusions about whether the gaze strategy adopted by BPD is driven by diagnostic features or due to some other factor such as social threat/vigilance it is important to systematically compare the gaze sampling in BPD individuals against the true distribution of the emotion-relevant information. This can reveal whether the eye movement strategy, such as eye avoidance, comes at the cost of sampling task-relevant information. An example of using such suboptimal strategies can be found in Chakravarthula, Tsank, and Eckstein 2021; Vision Research. Without such an analysis, we cannot definitively discount the threat avoidance hypothesis.
- Insufficient temporal resolution in gaze analysis. Emotion-priming paradigms often lead to transient, early-onset attentional biases. The use of aggregate gaze duration may mask such dynamic effects. Without a temporal decomposition of gaze behavior (e.g., first 500 ms vs. later viewing, or a fixation wise analysis), it is impossible to determine whether emotional primes influenced early orienting, subsequent disengagement, or later maintenance. Collapsing across time could dilute any subtle priming-driven modulations, especially given the short SOA and the neutral nature of the target faces. I also recommend looking into pupil size which is a continuous measure of arousal. There may be something interesting in the pupil size dynamics.
While the current study makes a valuable contribution to understanding affective processing in BPD a more temporally and spatially fine-grained analysis is warranted before concluding the absence of emotional priming in BPD. Such approaches would enhance the interpretability of the findings and may reveal subtle yet meaningful differences in automatic emotion processing.